# Fetal outcomes and associated factors of antepartum hemorrhage in Ethiopia: A systematic review and meta-analysis

Gemeda Wakgari Kitil[ID][1]*, Adamu Ambachew Shibabaw[2], Eyob Getachew[3], Amlaku Nigusie Yirsaw[3], Berihun Agegn Mengistie[4], Gebeyehu Lakew[3], Gebrehiwot Berie Mekonnen[5], Solomon Seyife Alemu[6], Firomsa Bekele[7], Lema Fikadu Wedajo[8], Addisalem Workie Demsash[9], Wubishet Gezimu[10], Mohammedamin Hajure Jarso[11], Geleta Nenko Dube[2], Fikadu Wake Butta[2], Alex Ayenew Chereka[ID][2]

1 Department of Midwifery, College of Health Science, Mattu University, Mattu, Ethiopia, 2 Department of Health Informatics, College of Health Science, Mattu University, Mattu, Ethiopia, 3 Department of Health Promotion and Health Behavior, Institute of Public Health, College of Medicine and Health Sciences, University of Gondar, Gondar, Ethiopia, 4 Department of General Midwifery, School of Midwifery, College of Medicine and Health Sciences, University of Gondar, Gondar, Ethiopia, 5 Department of Pediatrics and Child Health Nursing, College of Health Sciences, Debre Tabor University, Debre Tabor, Ethiopia, 6 Madda Walabu University, Shashamanne Campus, College of health science, Department of Midwifery, Shashamanne, Ethiopia, 7 School of Pharmacy, Institute of Health Science Wallaga University, Nekemte, Ethiopia, 8 Wallaga University, Institute of Health sciences, Department of Midwifery, Nekemte, Ethiopia, 9 Debre Berhan University, Asrat Woldeyes Health Science Campus, Health informatics Department, Debre Berhan, Ethiopia, 10 Department of Nursing, College of Health Science, Mattu University, Mattu, Ethiopia, 11 Madda walabu University, Shashemene Campus, Shashamanne, Ethiopia

* gemedawa425@gmail.com

## Abstract

### Background

Antepartum hemorrhage (APH) is a significant obstetric complication characterized by bleeding from the genital tract before the onset of labor. It poses serious risks to both maternal and fetal health, with potential outcomes including preterm birth, low birth weight, and increased perinatal mortality. In Ethiopia, where access to comprehensive prenatal care and emergency obstetric services may be limited, the impact of APH on fetal outcomes is a pressing concern. Despite the gravity of this condition, there is a lack of synthesized evidence on its prevalence and the associated risk factors specific to the Ethiopian context. This systematic review and meta-analysis aim to consolidate existing research on the fetal outcomes of APH and identify the key factors contributing to its incidence and severity in Ethiopia.

### Methods

Following the PRISMA checklist guidelines, this study concentrated on research undertaken in Ethiopia. Comprehensive searches across multiple databases Google Scholar, Google, Web of Science, and PubMed yielded six pertinent studies. Data were systematically gathered using a structured checklist and analyzed with STATA version 14. The Cochrane Q test and the I² statistic were utilized to assess heterogeneity. Additionally,

**Data availability statement:** The data and materials utilized in this research are accessible in the Supplementary Files provided.

**Funding:** The author(s) received no specific funding for this work.

**Competing interests:** The authors state that no personal or financial conflicts of interest might have affected the study's findings.

publication bias was examined using Egger's weighted regression, a funnel plot, and Begg's test.

## Results

This study reviewed 525 research articles that included a total of 22,914 participants. Our meta-analysis revealed that the pooled prevalence of perinatal mortality due to antepartum hemorrhage in Ethiopia is 21.79% (95% CI: 12.32–31.25). Key factors influencing perinatal mortality associated with antepartum hemorrhage included living in rural areas (OR = 1.62, 95% CI: 1.33–1.98), delays in seeking medical care for over 12 hours (OR = 5.43, 95% CI: 1.01–29.14), prematurity (OR = 9.00, 95% CI: 5.39–15.03), and experiencing severe vaginal bleeding prior to arrival at a medical facility (OR = 7.04, 95% CI: 2.14–23.13).

## Conclusion

The study reveals a high perinatal mortality rate of 21.79% associated with antepartum hemorrhage in Ethiopia. Contributing factors include rural residence, delays in seeking care, prematurity, and severe vaginal bleeding. To reduce mortality, improve rural healthcare, strengthen emergency systems, and enhance prenatal care. Public education on early intervention and specialized care for premature infants are also essential. Collaboration among healthcare providers, government, and NGOs is crucial for effective, sustainable solutions.

## 1. Introduction

Antepartum hemorrhage (APH), defined as bleeding from the genital tract after 28 weeks of gestation and before the onset of labor, is a significant obstetric complication. The leading causes of APH include placenta previa, placental abruption, and less common conditions, all of which can result in severe maternal and fetal morbidity and mortality [1,2].

Globally, APH complicates 2–5% of pregnancies and is associated with disproportionately high rates of adverse outcomes, contributing to approximately 20–30% of all perinatal deaths [2,3]. This burden is particularly severe in low-resource settings, where access to timely and quality maternal healthcare is limited. For example, in North India, APH has been linked to a perinatal mortality rate of 11.8% [4].

In Africa, the situation is even more concerning. The continent has some of the highest rates of perinatal mortality worldwide, with a particularly striking example being a study conducted at a northern Nigerian teaching hospital, which reported a stillbirth rate of 42.8% [5]

Ethiopia also faces a substantial burden of APH-related complications. A study conducted in the Tigray region revealed that 3.8% of all deliveries were complicated by APH, with perinatal mortality rates ranging from 15.8% to 30.6% [6–8]. The most common cause of APH in this study was abruptio placenta, accounting for 2.3% of cases [8]. In most instances, cesarean section is the preferred mode of delivery, accounting for 62.5% of cases [8].

The factors contributing to poor feto-maternal outcomes associated with APH in Ethiopia are multifaceted. These include limited antenatal care (ANC) coverage, rural residence, delays in accessing emergency care, advanced maternal age, previous cesarean section, grand multiparity, pregnancy-induced hypertension, and underlying medical conditions such as diabetes

and hypertension [6–15]. Fetal complications related to APH are particularly severe, including prematurity, low birth weight, anemia, sepsis, hypoxia, intrauterine fetal death, and stillbirth. Many cases require specialized neonatal care, including admission to a neonatal intensive care unit (NICU), to improve survival outcomes [1,8].

Efforts have been made in Ethiopia to address maternal and perinatal mortality. In 2013, the government introduced the Maternal Death Surveillance and Response (MDSR) system to systematically track and analyze maternal deaths, aiming to identify preventable causes and implement targeted interventions [16–18]. The MDSR system aligns with Sustainable Development Goal (SDG) 3.1, which seeks to reduce the maternal mortality ratio to below 70 per 100,000 live births by 2030 [19]. In conjunction with the MDSR system, Ethiopia has introduced several measures to reduce maternal mortality. These include maternity waiting room construction within health facilities and the provision of free transportation and maternity services [20–22]. However, despite these interventions, Ethiopia continues to face a persistently high maternal mortality ratio [23].

Many studies have examined fetal outcomes related to antepartum hemorrhage in Ethiopia, but there is still no comprehensive national estimate or consistent identification of key factors. Research findings are often inconclusive and vary by region, highlighting the need for a more unified analysis. This meta-analysis and review aim to fill this gap by providing a nationwide evaluation of fetal outcomes associated with antepartum hemorrhage in Ethiopia. The study will quantify perinatal mortality, identify important factors, and pinpoint gaps in current clinical practices. It also aims to improve healthcare policies and midwifery practices while raising awareness among healthcare providers and pregnant women. By combining existing evidence, this study will help advance maternal and child health across the country.

## 2. Methods and materials

### 2.1. Source of information and search strategy

To identify relevant studies for our analysis, we conducted an extensive search using two primary databases: the PROSPERO database and the Database of Abstracts of Reviews of Effects (DARE), both accessible via the UCSF Library. This comprehensive search aimed to capture a wide range of studies pertinent to our research objectives. Our search strategy was meticulously crafted following the guidelines outlined by the updated Preferred Reporting Items for Systematic Reviews and Meta-Analyses (PRISMA) [24]. These guidelines ensure a systematic and transparent approach to the review process, enhancing the reliability and reproducibility of our findings. The main objective of our study is to explore the various factors influencing fetal outcomes and investigate the associated factors of antepartum hemorrhage among women in Ethiopia. This includes a detailed examination of existing literature to identify trends, correlations, and gaps in the current knowledge base. The studies identified through our search were thoroughly reviewed and analyzed to extract relevant data (S1 Table).

To conduct a comprehensive literature review, we systematically searched multiple online databases, including Google Scholar, Google, Web of Science, and PubMed, from February 20, 2024, to May 30, 2024. Our search strategy combined keywords, free-text search queries, and Medical Subject Headings (MeSH). We utilized Boolean operators to combine various terms related to "prevalence," "magnitude," "fetal outcome," "feto-maternal outcome," "associated factors," "determinants," "antepartum hemorrhage," "placenta previa," and "placental abruption" with terms specifically associated with pregnant women. The structured search criteria were as follows: ("assessment," "prevalence," "associated factors," "determinants," "antepartum hemorrhage," "placenta previa," "placental abruption," "pregnant women") AND ("antepartum hemorrhage" OR "placenta previa" OR "placental abruption" OR "fetal outcomes and

associated factors of antepartum hemorrhage among women in Ethiopia"). This systematic approach ensured a thorough and structured examination of the literature within the specified period (S2 Table).

Our study sought to thoroughly investigate the range and determinants affecting fetal outcomes related to antepartum hemorrhage in Ethiopian women. We adopted a meticulous approach, systematically reviewing the titles, abstracts, and full texts of the chosen studies. To guarantee data accuracy, we adhered to the guidelines set forth by the Joanna Briggs Institute [25]. Four reviewers GWK, AAS, FWB, and AAC independently extracted data from every article. Any differences were resolved through joint discussions to ensure the reliability of our results.

This systematic review and meta-analysis lay the foundation for an in-depth examination of fetal outcomes and contributing factors of antepartum hemorrhage in Ethiopian women. The research seeks to offer a detailed understanding of existing literature and create a basis for valuable insights into the determinants of fetal outcomes in cases of antepartum hemorrhage in Ethiopia.

## 2.2. Eligibility criteria

Our systematic review's inclusion criteria were crafted to focus on original research articles examining the fetal outcomes and associated factors of antepartum hemorrhage among women in Ethiopia and its associated factors. We deemed all observational studies suitable for inclusion, regardless of their publication status (S3 Table). Our objective was to provide a wide-ranging and current synthesis of the literature, so we refrained from imposing limitations according to the year of publication. The review provides a current perspective on research, covering articles published until July 5, 2024. Strict exclusion criteria were applied to ensure accuracy and relevance: studies that did not investigate fetal outcomes and associated factors of antepartum hemorrhage among women in Ethiopia were excluded. Additionally, journals lacking abstracts or complete texts were disregarded to enable a thorough analysis of the selected studies. This methodological strategy attempted to preserve the level of quality and applicability of the analysis in the literature review (S4 Table).

This systematic review's primary goal was to look at cross-sectional and analytical cross-sectional studies that examined the fetal outcomes and contributing factors of antepartum hemorrhage in Ethiopian women. Specifically, only full-text English papers from readily accessible peer-reviewed journals were considered. Exclusion criteria included non-cross-sectional studies, studies done in languages other than English, national survey findings, conference reports, case studies, and expert opinions. Furthermore, this thorough evaluation omitted editorial reports, letters, reviews, and commentary.

In this meta-analysis and systematic review, the population of interest is women in Ethiopia who experience antepartum hemorrhage (P). The intervention involves analyzing and evaluating the various factors associated with fetal outcomes in these cases (I). The study compares different fetal outcomes—such as preterm birth, stillbirth, and low birth weight—based on the influence of factors like maternal health, medical interventions, and socio-economic conditions (C). The outcome of the review aims to identify key determinants of fetal outcomes and provide a wide-ranging understanding of their impact on pregnancies affected by antepartum hemorrhage (O).

## 2.3. Data extraction

The study's quality was assessed using the cross-sectional study quality checklist from the Joanna Briggs Institute (JBI) [25]. Four researchers (AAC, FWB, AAS, and GWK) compiled

data from the included studies into a computer spreadsheet, which a third researcher (LFW) later verified for consistency. Microsoft Excel was used by the two authors (GWK and AAC) to extract data and gather pertinent information using a predetermined checklist.

The initial phase involved merging search results from various databases and employing EndNote version 20.0 reference management software to identify and eliminate identical articles efficiently. Following this, study titles and abstracts underwent a thorough review to discard irrelevant entries. Finally, an in-depth analysis was conducted on the full-text versions of the remaining studies (S5 & S6 Table).

## 2.4. Data synthesis and analysis

The synthesis and analysis of the data process began with extracting the necessary data into a Microsoft Excel spreadsheet. Subsequently, the data were imported into STATA version 17 for in-depth analysis. The key findings were thoroughly described and summarized using forest plots, tables, and figures. A 95% confidence interval (CI) for a random-effects model was employed to generate an overall estimate of fetal outcomes and associated factors of antepartum hemorrhage among women in Ethiopia.

We utilized odds ratios with a 95% confidence interval to assess the link between fetal outcomes and the factors related to antepartum hemorrhage in women. Due to the heterogeneity observed among the studies, we opted for a random-effects model in our meta-analysis. This approach allowed for a comprehensive examination and understanding of the aggregated data.

To evaluate potential publication bias, we conducted Egger's regression tests and analyzed the figure of the funnel plot. Additionally, we assessed the differences in reported prevalence and outcomes of antepartum hemorrhage across studies using Cochran's Q test and $I^2$ statistics. tatistical significance was defined by a p-value of less than 0.05 from Cochran's Q test. The level of variation was indicated by the I2 statistic, which has a range of 0% to 100%: 0% indicates no variation, 25% minor variation, 50% moderate variation, and 75% or greater variation.

## 2.5. Outcome measurement

Perinatal mortality/outcome of antepartum hemorrhage (APH) refers to fetal or neonatal death and significant health complications due to bleeding episodes during pregnancy before labor onset. This includes stillbirth after 20 weeks of gestation, neonatal death within 28 days of birth, and adverse health effects in the fetus or neonate linked to APH, confirmed through clinical records and medical examinations.

In the first phase of our study, we examined how common perinatal mortality is in cases of antepartum hemorrhage and identified the factors that contribute to it. To collect this data, we have used a checklist that includes information on the study design, authors, response rate, publication year, sample size, study location, and the number of pregnant women who participated.

In the second part of our study, we looked at factors affecting perinatal mortality and outcomes related to antepartum hemorrhage. We used two-by-two tables to collect data and calculated the log odds ratio (OR) from the original research. Any disagreements between our reviewers, GWK and AAC, were resolved through additional responses and discussions. For articles that did not provide enough information, we reached out to the authors by email to get more details.

## 2.6. Quality assessment and appraisal

We employed a standardized instrument to recognize potential biases and variations in research findings to evaluate the studies' quality. A check for quality was done by four

different reviewers. We employed the Newcastle-Ottawa Scale (NOS), which is expressly intended to measure bias in observational research, to examine methodological concerns. The modified NOS scale was used to select only studies with a score of seven or higher (S7 Table).

### 2.7. Consent to publish and ethical approval

This systematic review followed the Preferred Reporting Items for Systematic Reviews and Meta-Analyses (PRISMA) guidelines with precision, emphasizing thoroughness and meticulous attention to detail. To address potential conflicts of interest and ensure fair representation, the included studies were subject to stringent review and approval. As a result, we have determined that ethical approval is not required, and the timing of participant recruitment and access to medical records are no longer pertinent concerns.

## 3. Results

### 3.1. Study selections

Initially, we collected 525 articles from several databases, encompassing a wide range of studies relevant to our research topic. After a meticulous screening process, we identified and removed 200 duplicate articles to ensure the uniqueness of our data set. We then scrutinized the titles and abstracts of the remaining articles, which led to the exclusion of 300 more papers that did not meet our inclusion criteria. Following this preliminary filtering, we conducted an in-depth review of the full texts of 25 publications to comprehensively assess their relevance and quality. Ultimately, we selected six specific studies that met all our stringent criteria for inclusion in our meta-analysis and systematic review, ensuring a robust and focused analysis of the available evidence (Fig 1).

### 3.2. Characteristics included in the reviewed research

Data from six studies carried out in Ethiopia were examined by our thorough meta-analysis and systematic review, each providing significant insights into fetal outcomes of antepartum hemorrhage among women in the country. The combined sample size across all studies was

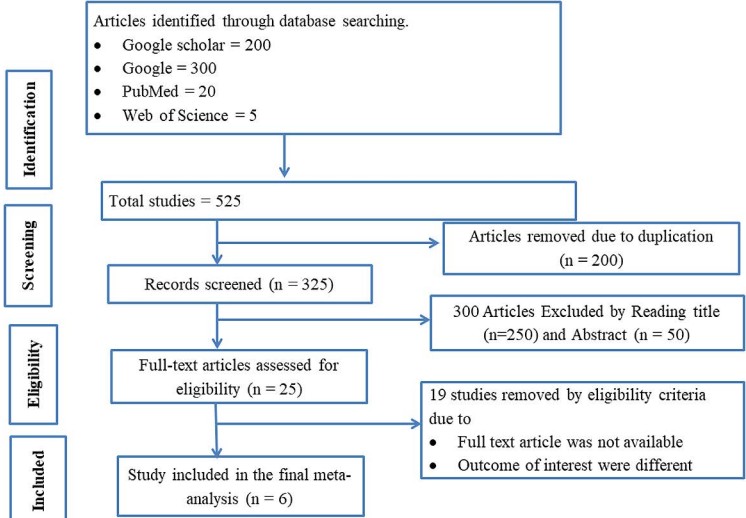

**Fig 1. A PRISMA method for selecting studies in a systematic review and meta-analysis.**

22,914 deliveries, with participant numbers ranging from 377 to 9,643 per study [9,10]. Notably, the study conducted in Addis Ababa had the largest sample size with 9,643 participants [9], while the study from Jimma in the Oromia region had the smallest sample size with 377 participants [10]. All studies used a cross-sectional design for data collection.

Geographically, these studies were distributed across various regions of Ethiopia. Specifically, four studies were carried out in the Oromia region [7–10], one in the Amhara region [13], and one in the Tigray region [6] (Table 1).

### 3.3. Pooled prevalence of perinatal mortality/outcome of antepartum hemorrhage in Ethiopia

Our meta-analysis determined that the pooled prevalence of perinatal mortality associated with antepartum hemorrhage in Ethiopia is 21.79% (95% CI: 12.32–31.25). This figure underscores the significant risk posed by antepartum hemorrhage to fetal outcomes in the country.

Among the analyzed studies, the highest prevalence of perinatal mortality due to antepartum hemorrhage was reported in the Oromia region, particularly at the Jimma Town Public Health Institution, with a rate of 30.6% (95% CI: 4.53–56.67) [7]. In contrast, the study conducted in Addis Ababa reported the lowest prevalence rate of 15.8% [9].

The analysis revealed minimal heterogeneity between the involved studies, indicated by a p-value of 0.957 and an I2 value of 0.0% (Fig 2).

### 3.4. Publication bias

We assessed publication bias through Egger's regression test and by visually inspecting a funnel plot. Initially, the funnel plot showed an uneven distribution. However, Egger's test result (P = 0.339) did not indicate statistical significance. Therefore, our analysis does not reveal substantial evidence of publication bias in the reviewed research articles. This conclusion is corroborated by Egger's regression test findings (Fig 3).

### 3.5. Factors associated with perinatal mortality/outcome of antepartum hemorrhage in Ethiopia

The purpose of our meta-analysis was to determine the variables affecting perinatal mortality and outcomes of antepartum hemorrhage in Ethiopia by synthesizing data from six studies. This analysis employed the command 'metan logor selogor, xlab(0.1, 1, 10) label(namevar = authors) by (factors) random texts(180) eform,' enabling a thorough assessment of the odds ratios' cumulative effects.

**Table 1. Overview of the six studies included in the meta-analysis on the Fetal Outcomes and associated factors of Antepartum Hemorrhage among Women in Ethiopia.**

| Authors | Year | Region | Study area | Study design | Study population | Quality score | Sample Size | Response rate | magnitude of perinatal mortality |
|---|---|---|---|---|---|---|---|---|---|
| Asefa et al [9] | 2020 | Addis Ababa | Addis Ababa | Cross-sectional | newborns | 8 | 9643 | 100.00% | 15.8 |
| Hailu et al [6] | 2023 | Tigray | Ayder | Cross-sectional | newborns | 8 | 5368 | 100.00% | 22.5 |
| Kassie et al [8] | 2019 | Oromia | Illu Ababora Zone | Cross-sectional | newborns | 8 | 3224 | 100.00% | 22.8 |
| Zegeye et al [13] | 2024 | Amhara | Awi zone | Cross-sectional | newborns | 9 | 448 | 100.00% | 17.4 |
| Chufamo et al [7] | 2015 | Oromia | Jimma | Cross-sectional | newborns | 8 | 3854 | 100.00% | 30.6 |
| Gelan et al [10] | 2022 | Oromia | Jimma | Cross-sectional | newborns | 9 | 377 | 98.00% | 26.5 |

Kassie et al [8] = URI: http://10.140.5.162//handle/123456789/4081.

The results revealed several key factors significantly associated with perinatal mortality and adverse outcomes in cases of antepartum hemorrhage. Rural residence was found to significantly increase the risk (OR = 1.62, 95% CI: 1.33–1.98). Additionally, delays in seeking care for more than 12 hours were associated with a substantially higher risk (OR = 5.43, 95% CI: 1.01–29.14). Prematurity emerged as a critical factor, with a notably elevated risk (OR = 9.00, 95% CI: 5.39–15.03). The severity of vaginal bleeding before arrival also played a significant role, with moderate bleeding associated with an OR of 7.04 (95% CI: 2.14–23.13) and severe bleeding with an OR of 4.26 (95% CI: 1.22–14.89). These findings underscore the importance

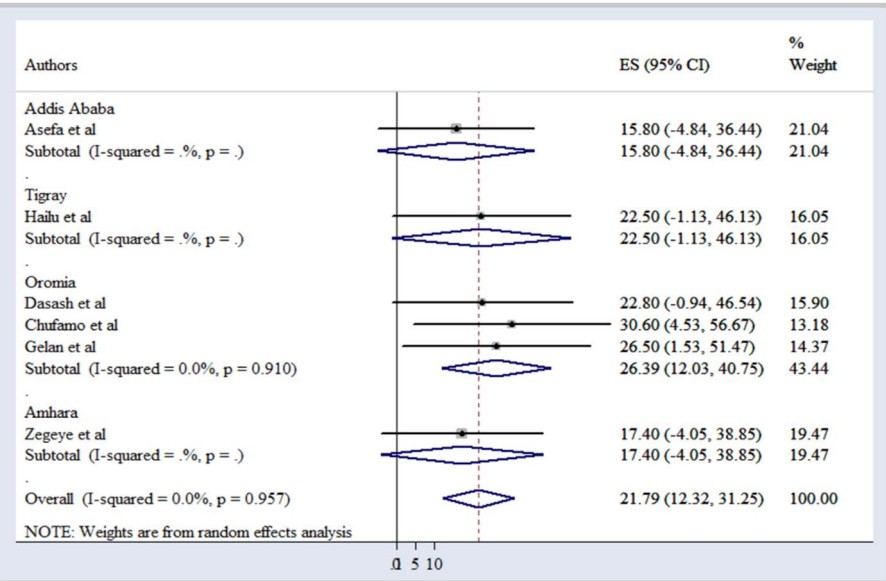

**Fig 2. A forest plot of the pooled prevalence of perinatal mortality/outcome of antepartum hemorrhage in Ethiopia.**

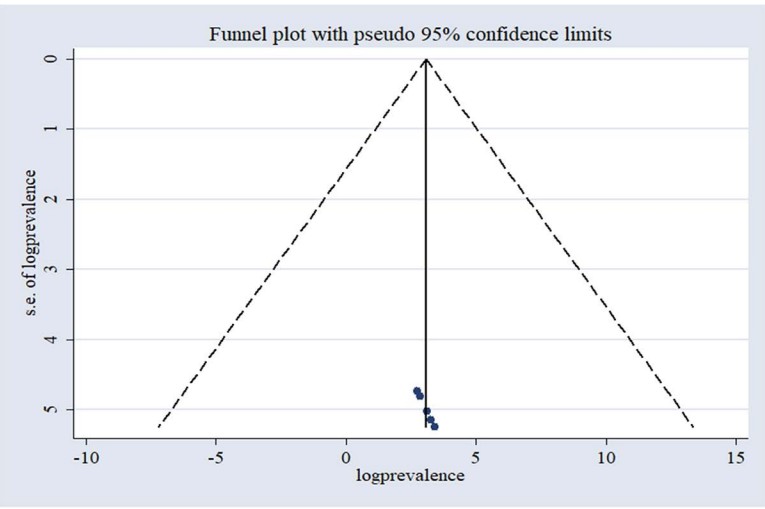

**Fig 3. A funnel plot of the included studies, assessing publication bias.**

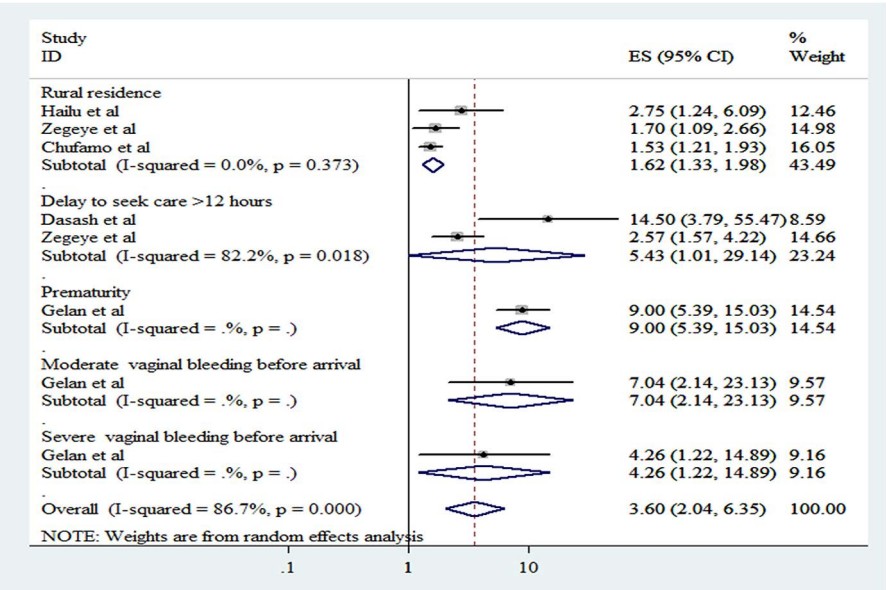

**Fig 4. Forest plot of Factors associated with perinatal mortality/outcome of antepartum hemorrhage in Ethiopia.**

of timely medical intervention and targeted strategies to improve perinatal outcomes in the context of antepartum hemorrhage (Fig 4).

## 4. Discussion

Our meta-analysis provides a comprehensive overview of the perinatal mortality associated with antepartum hemorrhage in Ethiopia, highlighting several critical factors influencing outcomes. The pooled prevalence of perinatal mortality at 21.79% (95% CI: 12.32–31.25) underscores a significant concern within this population, suggesting that antepartum hemorrhage is a substantial contributor to adverse perinatal outcomes. The meta-analysis and systematic review revealed that our study's finding of a pooled prevalence of perinatal mortality at 21.79% (95% CI: 12.32–31.25) is lower than the 42.8% stillbirth rate reported in a study conducted at a northern Nigerian teaching hospital [5].

However, our study's prevalence of perinatal mortality is notably higher than the 11.84% rate reported in a study from North India that also focused on antepartum hemorrhage (APH) and its outcomes [4]. This discrepancy may be due to limited access to quality healthcare, especially in remote areas, which can delay timely interventions during pregnancy and childbirth. Socioeconomic disparities, such as poverty and inadequate maternal nutrition, likely contribute to poorer outcomes. Additionally, cultural practices that delay seeking medical care and differences in the effectiveness of health systems may further exacerbate mortality rates.

From our analysis, we identified several key factors. Women living in rural areas faced a higher risk of perinatal mortality and adverse fetal outcomes compared to their urban counterparts (OR = 1.62, 95% CI: 1.33–1.98). This result is in line with research done at Shands Hospital in Florida [26]. The increased risk in rural areas may be due to factors such as greater distances to healthcare facilities, lower socioeconomic status, limited media access, and less frequent antenatal care visits during pregnancy.

This study found that women who delayed seeking medical care for more than 12 hours had a significantly higher likelihood of poor fetal outcomes and perinatal mortality compared

to those who sought care earlier (OR = 5.43, 95% CI: 1.01–29.14). This finding is supported by a study on maternal and newborn healthcare-seeking in low- and middle-income countries [27]. The increased risk might be due to limited awareness of pregnancy warning signs, flaws in the early referral process, insufficient community education, and poor transportation access. These factors highlight the critical impact of timely medical intervention in managing antepartum hemorrhage and underscore the need for improved community education and infrastructure to facilitate faster access to care.

Prematurity was another significant factor affecting perinatal mortality; newborns born prematurely were 9 times more likely to experience perinatal mortality compared to those born at term or later (OR = 9.00, 95% CI: 5.39–15.03). The strong association between prematurity and perinatal mortality underscores the importance of prenatal care and monitoring to prevent preterm births and manage complications promptly.

Additionally, the severity of vaginal bleeding before arrival at a healthcare facility was identified as a significant predictor of perinatal outcomes. Specifically, moderate and severe vaginal bleeding was linked to increased risks of perinatal mortality or adverse fetal outcomes. Moderate bleeding was associated with a fourfold increase in risk (OR = 4.26, 95% CI: 1.22–14.89), while severe bleeding was associated with a sevenfold increase in risk (OR = 7.04, 95% CI: 2.14–23.13). This was supported by a previous study [28]. The increased risk of perinatal mortality or adverse fetal outcomes associated with moderate to severe vaginal bleeding can be attributed to its indication of serious conditions such as placental abruption or previa, which compromise fetal health. These findings highlight the critical need for prompt medical intervention to manage these risks effectively and improve outcomes for both the mother and fetus.

### 4.1.  Strengths and limitations

This meta-analysis and systematic review have several strengths. It provides a thorough overview by combining data from multiple studies, which enhances the reliability of the findings. By identifying key factors such as rural residence, delayed medical care, prematurity, and the severity of vaginal bleeding, the review offers valuable insights for improving perinatal health in Ethiopia. The study's focus on Ethiopia ensures that the recommendations are relevant to the local context.

However, there are also limitations. The quality of the included studies varies, which may impact the consistency of the results. Additionally, there might be gaps in data on certain factors or regional differences that could affect the completeness of the findings. There is also a potential for publication bias, as studies with significant results are more likely to be published. Finally, while the review is specific to Ethiopia, its conclusions may not fully apply to other regions with different healthcare systems or socio-economic conditions.

### 4.2.  Conclusion

Our comprehensive review and meta-analysis concluded that antepartum hemorrhage significantly contributes to perinatal mortality in Ethiopia, with a concerning pooled prevalence of 21.79%. This analysis identifies several key factors influencing perinatal mortality, including rural residence, delayed medical care, prematurity, and the severity of vaginal bleeding. Addressing these factors is essential for improving perinatal outcomes in this population.

### 4.3.  Recommendation

To reduce the impact of antepartum hemorrhage and associated perinatal mortality, efforts should focus on improving healthcare access in rural areas by strengthening infrastructure

and resources to ensure timely medical care. Comprehensive educational programs for expectant mothers and healthcare providers should emphasize early recognition of antepartum hemorrhage and prompt intervention. Enhancing prenatal care services to include regular monitoring and management of high-risk conditions is essential. Strategies promoting early detection and intervention for high-risk pregnancies should be prioritized, alongside investments in rural health infrastructure to upgrade medical facilities and training programs for healthcare professionals. Additionally, supporting ongoing research into the causes and management of antepartum hemorrhage will aid in developing effective prevention and treatment strategies.

## Supporting Information

**S1 Table. A PRISMA 2020 checklist for Systematic Review and Meta-Analysis on Perinatal Mortality/Outcome of Antepartum Hemorrhage in Ethiopia.**
(DOCX)

**S2 Table. Search strategy.**
(DOCX)

**S3 Table. Studies Included in the Systematic Review and Meta-Analysis.**
(DOCX)

**S4 Table. Reasons for Exclusion of Studies Identified During the Literature Search.**
(DOCX)

**S5 & 6 Table. Supporting tables of all data extracted from the primary research sources for the systematic review and meta-analysis.**
(DOCX)

**S7 Table. Meta-Analysis and Systematic Review Quality Assessment.**
(DOCX)

## Acknowledgments

We would like to thank the investigators whose initial research was incorporated into this systematic review and meta-analysis. Their invaluable contributions provided the foundational data for this research.

## Author contributions

**Conceptualization:** Gemeda Wakgari Kitil, Adamu Ambachew Shibabaw, Eyob Getachew, Gebrehiwot Berie mekonnen, Solomon Seyife Alemu, Firomsa Bekele, Lema Fikadu Wedajo, Addisalem Workie Demsash, Wubishet Gezimu, Fikadu Wake Butta, Alex Ayenew Chereka.

**Data curation:** Gemeda Wakgari Kitil, Adamu Ambachew Shibabaw, Amlaku Nigusie Yirsaw, Berihun Agegn Mengistie Mengistie, Gebrehiwot Berie mekonnen, Solomon Seyife Alemu, Firomsa Bekele, Lema Fikadu Wedajo, Addisalem Workie Demsash, Wubishet Gezimu, Alex Ayenew Chereka.

**Formal analysis:** Gemeda Wakgari Kitil.

**Funding acquisition:** Gemeda Wakgari Kitil.

**Investigation:** Gemeda Wakgari Kitil.

**Methodology:** Gemeda Wakgari Kitil.

**Project administration:** Gemeda Wakgari Kitil.

**Resources:** Gemeda Wakgari Kitil.

**Software:** Gemeda Wakgari Kitil.

**Supervision:** Gemeda Wakgari Kitil, Adamu Ambachew Shibabaw, Eyob Getachew, Amlaku Nigusie Yirsaw, Berihun Agegn Mengistie Mengistie, Gebeyehu Lakew, Gebrehiwot Berie mekonnen, Solomon Seyife Alemu, Firomsa Bekele, Lema Fikadu Wedajo, Wubishet Gezimu, Mohammedamin Hajure Jarso, Geleta Nenko Dube, Fikadu Wake Butta, Alex Ayenew Chereka.

**Validation:** Gemeda Wakgari Kitil, Eyob Getachew, Amlaku Nigusie Yirsaw, Gebrehiwot Berie mekonnen, Firomsa Bekele, Addisalem Workie Demsash, Wubishet Gezimu, Geleta Nenko Dube, Alex Ayenew Chereka.

**Visualization:** Gemeda Wakgari Kitil, Adamu Ambachew Shibabaw, Eyob Getachew, Amlaku Nigusie Yirsaw, Berihun Agegn Mengistie Mengistie, Gebeyehu Lakew, Solomon Seyife Alemu, Firomsa Bekele, Lema Fikadu Wedajo, Addisalem Workie Demsash, Mohammedamin Hajure Jarso, Geleta Nenko Dube, Fikadu Wake Butta, Alex Ayenew Chereka.

**Writing – original draft:** Gemeda Wakgari Kitil.

**Writing – review & editing:** Gemeda Wakgari Kitil, Adamu Ambachew Shibabaw, Eyob Getachew, Gebeyehu Lakew, Firomsa Bekele, Lema Fikadu Wedajo, Addisalem Workie Demsash, Fikadu Wake Butta, Alex Ayenew Chereka.

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
