## [Decision Letter · Decision Letter 0]

9 Dec 2024

Dear Dr. Kitil,

Thank you for submitting your manuscript to PLOS ONE. After careful consideration, we feel that it has merit but does not fully meet PLOS ONE’s publication criteria as it currently stands. Therefore, we invite you to submit a revised version of the manuscript that addresses the points raised during the review process.

**ACADEMIC EDITOR: Please respond to all reviewers comments**

We look forward to receiving your revised manuscript.

Kind regards,

Ahmed Mohamed Maged, MD

Academic Editor

PLOS ONE

 Journal Requirements: When submitting your revision, we need you to address these additional requirements. 1. Please ensure that your manuscript meets PLOS ONE's style requirements, including those for file naming. The PLOS ONE style templates can be found at https://journals.plos.org/plosone/s/file?id=wjVg/PLOSOne_formatting_sample_main_body.pdf and https://journals.plos.org/plosone/s/file?id=ba62/PLOSOne_formatting_sample_title_authors_affiliations.pdf 2. As required by our policy on Data Availability, please ensure your manuscript or supplementary information includes the following:  A numbered table of all studies identified in the literature search, including those that were excluded from the analyses.   For every excluded study, the table should list the reason(s) for exclusion.   If any of the included studies are unpublished, include a link (URL) to the primary source or detailed information about how the content can be accessed.  A table of all data extracted from the primary research sources for the systematic review and/or meta-analysis. The table must include the following information for each study:  Name of data extractors and date of data extraction  Confirmation that the study was eligible to be included in the review.   All data extracted from each study for the reported systematic review and/or meta-analysis that would be needed to replicate your analyses.  If data or supporting information were obtained from another source (e.g. correspondence with the author of the original research article), please provide the source of data and dates on which the data/information were obtained by your research group.  If applicable for your analysis, a table showing the completed risk of bias and quality/certainty assessments for each study or outcome.  Please ensure this is provided for each domain or parameter assessed. For example, if you used the Cochrane risk-of-bias tool for randomized trials, provide answers to each of the signalling questions for each study. If you used GRADE to assess certainty of evidence, provide judgements about each of the quality of evidence factor. This should be provided for each outcome.   An explanation of how missing data were handled.  This information can be included in the main text, supplementary information, or relevant data repository. Please note that providing these underlying data is a requirement for publication in this journal, and if these data are not provided your manuscript might be rejected.

Reviewers' comments:

**Comments to the Author**

1. Is the manuscript technically sound, and do the data support the conclusions?

Reviewer #1: Yes

Reviewer #2: Yes

2. Has the statistical analysis been performed appropriately and rigorously?

Reviewer #1: Yes

Reviewer #2: Yes

3. Have the authors made all data underlying the findings in their manuscript fully available?

Reviewer #1: Yes

Reviewer #2: Yes

4. Is the manuscript presented in an intelligible fashion and written in standard English?

Reviewer #1: Yes

Reviewer #2: Yes

Reviewer #1: Dear Editor, Thanks for your invitation to review this interesting paper. The paper adds a value on clinical practice and suitable for publication after addressing below comments

1. The PRSIMA version used should be mentioned. I highly recommend the author to use PRISMA 2020(Page MJ, McKenzie JE, Bossuyt PM, Boutron I, Hoffmann TC, Mulrow CD, et al. The PRISMA 2020 statement: an updated guideline for reporting systematic reviews. BMJ 2021;372:n71. doi: 10.1136/bmj.n71)

2. The PubMed and Medline are same database while Google cannot be a data base

3. In Table 1, Better to include CI for your primary outcome

4. Better to write a recommendation with a single paragraph

5. The public and patient involvement sub title should be removed as it is unnecessary as per PLoS One guideline

Reviewer #2: Thank you for your comments.

I want to ask if the author updated the search criteria because I found more articles related to this topic.

The introduction is so long and needs to focus on the topic.

Does the author provide a publication bias for six studies?

where the risk of bias assessment.

**Do you want your identity to be public for this peer review?** For information about this choice, including consent withdrawal, please see our Privacy Policy

Reviewer #1: No

Reviewer #2: No

---

## [Author Response · Author response to Decision Letter 1]

23 Dec 2024

Dear Editor(s),

We hope this message finds you well.

We are pleased to submit the revised version of our manuscript titled "Fetal Outcomes and Associated Factors of Antepartum Hemorrhage in Ethiopia: A Systematic Review and Meta-Analysis" (PONE-D-24-38076) to PLOS ONE.

We sincerely appreciate your invaluable support throughout the review process. The constructive feedback and detailed insights provided by the editors and reviewers have greatly enhanced the clarity, rigor, and overall quality of our manuscript.

Thank you for considering our revised submission. We look forward to your feedback and remain available to address any further revisions or clarifications.

Best regards,

Gemeda Wakgari Kitil

Corresponding Author

gemedawa425@gmail.com

1. Point-by-point response letter to Editor

Ahmed Mohamed Maged, MD; Academic Editor of PLOS ONE

I hope this message finds you well. On behalf of my co-authors and myself, I would like to express our sincere gratitude for your invaluable efforts in handling our manuscript, " Fetal Outcomes and Associated Factors of Antepartum Hemorrhage in Ethiopia: A Systematic Review and Meta-Analysis " Your comments and suggestions have been incredibly helpful in refining our work. We greatly appreciate the time and expertise you dedicated to providing thorough and thoughtful feedback. Thank you once again for your guidance and support throughout the review process. Here is a point-by-point response addressing the changes made to your suggestions and comments:

Editor: Please submit your revised manuscript by Jan 23, 2025 11:59 PM. Please include the rebuttal letter for the response of the reviewer and editor, marked, and unmarked document when submitting your revised manuscript.

Authors: We sincerely appreciate your insightful comment. We have taken your feedback seriously and diligently addressed the concern you mentioned. Additionally, we have ensured that all necessary documents, as per your recommendation, have been submitted along with the revised manuscript within a given time. Thank you for guiding us through this process.

Editor: When submitting your revision, we need you to address these additional requirements.

Authors: Thank you for providing us with the additional requirements for our manuscript submission to PLOS ONE. We appreciate your thorough review and guidance to ensure compliance with the journal's style and submission standards. Below, we address each of the points raised:

Editor comment-1: Please ensure that your manuscript meets PLOS ONE's style requirements, including those for file naming

Authors: Thank you for your review and specific comments. We have carefully reviewed the PLOS ONE style requirements outlined in the provided templates and ensured that our manuscript adheres to these guidelines for file naming and formatting. Any necessary adjustments have been made to align our manuscript with the specified style. Please review the revised manuscript.

Editor comment-2: As required by our policy on Data Availability, please ensure your manuscript or supplementary information includes the following:

Authors: Thank you for your valuable feedback and for highlighting the data availability requirements. We have made the necessary revisions to ensure compliance with the policy. Please find the following updates and additions in the revised manuscript and supplementary materials:

1. Numbered Table of All Studies Identified in the Literature Search

We have included a numbered table listing all studies identified during our literature search, including those that were excluded from the analyses. Each excluded study is accompanied by a clear explanation of the reason for exclusion.

2. Unpublished Studies and Primary Source Access

We have included details about unpublished studies used in our analysis. These studies are clearly marked in the table 1, and access to their primary source has been ensured. For one such study, we have provided the following URL for reference: http://10.140.5.162//handle/123456789/4081.

3. Table of Data Extracted from Primary Research Sources

A comprehensive table has been added, detailing all the data extracted from the primary sources for our systematic review and meta-analysis. This includes:

o Name of the data extractors and the date of data extraction.

o Confirmation that each study met the eligibility criteria for inclusion.

o All data extracted from each study, enabling replication of our analyses.

4. Source of Data for Supporting Information

Where applicable, we have indicated if data or supporting information was obtained from external sources, such as correspondence with the authors of the original research articles. The specific sources and the dates on which the data were obtained are provided in the supplementary materials.

5. Risk of Bias and Quality/Certainty Assessments

A table summarizing the completed risk of bias and quality/certainty assessments for each study or outcome has been included. If we used the Cochrane Risk-of-Bias tool, we have provided answers to each of the signalling questions for each study. Additionally, if GRADE was used for assessing the certainty of evidence, judgments about each quality factor for every outcome have been provided.

6. Handling of Missing Data

We have included a detailed explanation of how missing data were handled in our analysis. This description is outlined in the methods section of the manuscript, ensuring transparency in our approach to missing data.

Point-by-point response letter to Reviewer-1

Authors: Thank you for your valuable feedback on our manuscript, "Fetal Outcomes and Associated Factors of Antepartum Hemorrhage in Ethiopia: A Systematic Review and Meta-Analysis." We greatly appreciate the time and effort you have invested in providing insightful comments and suggestions. We have thoroughly considered your feedback and made the necessary revisions. Below, we address each of the points raised:

Reviewer-1 Comment-1: The PRSIMA version used should be mentioned. I highly recommend the author to use PRISMA 2020(Page MJ, McKenzie JE, Bossuyt PM, Boutron I, Hoffmann TC, Mulrow CD, et al. The PRISMA 2020 statement: an updated guideline for reporting systematic reviews. BMJ 2021;372:n71. doi: 10.1136/bmj.n71)

Authors: Thank you for your valuable suggestion. We acknowledge the importance of adhering to the updated guidelines for reporting systematic reviews. In response to your comment, we have revised the manuscript to specify that we utilized the PRISMA 2020 checklist and flow diagram. We invite to review the updated manuscript.

Reviewer-1 comment-2: The PubMed and Medline are same database while Google cannot be a data base.

Authors: Thank you for pointing this out. We agree that PubMed is a platform that provides access to the MEDLINE database, among other resources. In our manuscript, we have revised the text to clarify this distinction and ensure accurate terminology.

Regarding Google, we acknowledge that it is not a database but rather a search engine. We have revised our methodology to specify that searches using Google were conducted to identify grey literature and additional sources outside traditional databases. We invited you to review our revised manuscript for further information.

Reviewer-1 comment-3. In Table 1, Better to include CI for your primary outcome

Authors: Thank you for your insightful suggestion. We agree that including the confidence intervals (CIs) for the primary outcome would enhance the clarity and interpretability of the results. However, upon further review of the articles included in our study, we found that most studies reported fetal outcomes collectively (e.g., preterm birth, low birth weight, and stillbirth) without providing CIs for individual outcomes. Due to this limitation in the available data, it is challenging to compute or extract the required CIs for all relevant outcomes. Nevertheless, we have ensured that the available effect sizes and other key statistics are presented clearly to maintain the integrity and transparency of our findings.

Reviewer-1 comment-4: Better to write a recommendation with a single paragraph.

Authors: Thank you for your suggestion. We have revised the recommendation section to condense it into a single, cohesive paragraph. This ensures clarity and conciseness while maintaining the key points. For further confirmation please refer to the revised tracked manuscript, for details.

Reviewer-1 comment-5: The public and patient involvement sub title should be removed as it is unnecessary as per PLoS One guideline

Authors: Thank you for pointing this out. We have reviewed the PLOS ONE guidelines and agree that the "Public and Patient Involvement" section is not required. In response, we have removed this subtitle from the manuscript to ensure compliance with the journal’s guidelines.

Point-by-point response letter to Reviewer-2

Authors: Thank you for your valuable feedback on our manuscript entitled " Fetal Outcomes and Associated Factors of Antepartum Hemorrhage in Ethiopia: A Systematic Review and Meta-Analysis." We appreciate the time you have taken to provide thoughtful comments and suggestions for improving our work. We have carefully considered your feedback and made the necessary revisions to address your comments. Below, we provide our responses to each of the points raised:

Reviewer-2 Comment-1 I want to ask if the author updated the search criteria because I found more articles related to this topic.

Authors: Thank you for your valuable observation. We have thoroughly updated the search criteria during the revision process to ensure the inclusion of the most recent and relevant studies on this topic. While we identified additional articles related to the subject, they were not included in our analysis because their outcomes of interest differed from the focus of our study. We prioritized studies that aligned closely with our research objectives and eligibility criteria.

Reviewer-2 Comment-2: The introduction is so long and needs to focus on the topic.

Authors: We appreciate your feedback. In response, we have revised the introduction to make it more concise and focused on the main topic of the manuscript. Unnecessary details were removed, and we have streamlined the content to provide a clear context for the study without excessive elaboration. See the revised manuscript.

Reviewer-2 Comment-3: Does the author provide a publication bias for six studies?

Authors: Thank you for raising this point. We have now addressed the issue of publication bias in the manuscript. In the revised version, we have included an assessment of publication bias for the six studies included in our analysis. We utilized funnel plots and Egger’s test to assess the presence of publication bias, and the results are discussed in the revised manuscript.

Reviewer-2 Comment-4: where the risk of bias assessment?

Authors: Thank you for your comment. The risk of bias assessment is detailed in section 2.6, titled "Quality Assessment and Appraisal." In this section, we describe how we employed a standardized instrument to identify potential biases and variations in research findings, thereby evaluating the quality of the included studies. A quality check was conducted by four independent reviewers. We used the Newcastle-Ottawa Scale (NOS), a tool specifically designed to assess bias in observational research, to evaluate the methodological quality of the studies. The modified NOS scale was used to include only studies with a score of seven or higher, ensuring that only studies with a low risk of bias were selected for inclusion in our analysis.

---

## [Decision Letter · Decision Letter 1]

4 Feb 2025

Fetal Outcomes and Associated Factors of Antepartum Hemorrhage in Ethiopia: A Systematic Review and Meta-Analysis

PONE-D-24-38076R1

Dear Dr. Kitil,

We’re pleased to inform you that your manuscript has been judged scientifically suitable for publication and will be formally accepted for publication once it meets all outstanding technical requirements.

Kind regards,

Ahmed Mohamed Maged, MD

Academic Editor

PLOS ONE

Additional Editor Comments (optional):

Reviewers' comments:

Reviewer's Responses to Questions

**Comments to the Author**

Reviewer #1: All comments have been addressed

2. Is the manuscript technically sound, and do the data support the conclusions?

Reviewer #1: Yes

3. Has the statistical analysis been performed appropriately and rigorously?

Reviewer #1: Yes

4. Have the authors made all data underlying the findings in their manuscript fully available?

Reviewer #1: Yes

5. Is the manuscript presented in an intelligible fashion and written in standard English?

Reviewer #1: Yes

Reviewer #1: The paper is among high quality. It have a stong value in low resource settings. Therefore, reccomend to accept the paper .

**Do you want your identity to be public for this peer review?** For information about this choice, including consent withdrawal, please see our Privacy Policy

Reviewer #1: No

---

## [Editor Report · Acceptance letter]

PONE-D-24-38076R1

PLOS ONE

Dear Dr. Kitil,

I'm pleased to inform you that your manuscript has been deemed suitable for publication in PLOS ONE. Congratulations! Your manuscript is now being handed over to our production team.

Kind regards,

on behalf of

Professor Ahmed Mohamed Maged

Academic Editor

PLOS ONE